# How Are Landscapes under Agroecological Transition Perceived and Appreciated? A Belgian Case Study

**Fanny Boeraeve** [1,*]**, Marc Dufrêne** [1] 🟢**, Nicolas Dendoncker** [2]**, Amandine Dupire** [1,3] **and Grégory Mahy** [1]

[1] Biodiversity and Landscape, TERRA Teaching and Research Center, Gembloux Agro-Bio Tech, University of Liege, Passage des Déportés 2, 5030 Gembloux, Belgium; marc.dufrene@uliege.be (M.D.); amandine.dupire@gmail.com (A.D.); g.mahy@uliege.be (G.M.)

[2] Department of Geography, Institute Transitions, University of Namur, Rue de Bruxelles 61, 5000 Namur, Belgium; nicolas.dendoncker@unamur.be

[3] Architecture and Planning, Urba Folia, Avenue de Canteleu 63, 59650 Villeneuve-d'Ascq, France

**\*** Correspondence: f.boeraeve@uliege.be

**Abstract:** An increasing number of agricultural transition initiatives are taking place, seeking more autonomy and resilience on the farms. This undeniably reshapes the landscape and the delivery of ecosystem services (ES). To date, little research includes the knowledge and perceptions of local communities on rural landscapes in agricultural transition. Yet, farmers shape the landscape and ES delivery, and local inhabitants are directly impacted. The present work aims at assessing the extent to which locals (local inhabitants and farmers) appreciate and view landscapes undergoing agricultural transitions. To do so, questionnaires were submitted to locals enquiring about appreciation and ES perceptions of transitioning landscapes. These landscapes were shown in manipulated photographs simulating an agroecological landscape, a conventional agriculture landscape, and landscapes including each isolated agroecological practice (resulting in six 'scenarios'). In order to put locals' perceptions in perspective, the same questionnaire was submitted to 'ES experts', and ES perceptions were compared to field-based ES measurements in agroecological and conventional parcels of the same study region. The results show that locals and ES experts appreciate and perceive these scenarios similarly. The agroecological scenario was seen as the most appreciated and the one delivering the most ES, while the conventional one was the least appreciated and seen as the one delivering the least ES. These perceptions of ES delivery partially correspond to the ES field measurements, which showed a similar productivity within agroecological and conventional parcels and more regulating ES in agroecological parcels. We discuss how our results call for the assessment of the multi-performance of agricultural systems in terms of ES rather than focusing on yield only, and how future research addressing agroecological transition should rely on integrated valuations and mixed methods to better embrace the complexity of such transitioning systems.

**Keywords:** ecosystem services; sustainable agriculture; stakeholders; consultation; perception; landscape; integrated valuation; mixed methods research

---

## 1. Introduction

Scientific literature abounds to warn about the environmental, social, and economic limitations of the current intensive agricultural model [1,2]. In response to these concerns, agroecology is being promoted as a promising concept [3,4]. In a recent review, Hatt et al. [5] defined agroecology as the application of ecological practices as well as the consideration of socio-economic dimensions for sustainable food systems. Agroecological practices rely on the hypothesis that modifying the

agroecosystem by restoring the agro-landscape structure and processes redefines delivery of ecosystem services (ES), some of which are crucial to the long-term performance of agriculture (e.g., natural pest control and natural soil fertility) [6–8].

Agroecology is largely and increasingly embraced by the scientific community [4,5,9,10] and also by farmers themselves to meet more resilience and autonomy [11]. The increasing number of farms shifting to organic farming [12], implementing Agro-Environmental Measures [13], putting conservation agriculture into practice [14], and organizing short supply chains [15] is illustrative of these emerging interests.

In the Western part of the Hainaut Province in Belgium, a core group of innovating farmers spontaneously changed their agricultural practices (e.g., feed autonomy, no-till agriculture, organic farming, etc.). While transitioning towards these innovative practices, the challenge for farmers lies in the numerous uncertainties related to the complex nature of agroecosystems, in which ecological processes and ES form an intricate network that is often unpredictable, not fully understood [16], and specific to each production site [17,18]. To tackle the challenge, these farmers have created a network entitled the 'innovating farms network' aiming at providing a 'safe learning space' where they can exchange knowledge and experiences [19,20]. As this network of farmers is gaining momentum, parts of the landscape are gradually undergoing a shift from the typical simple and homogenous landscapes of conventional agriculture in Western Europe to a more complex and heterogeneous landscape.

Rural landscapes represent the place where many people live, recreate [21], and with which they create a feeling of identity and belonging [22]. They also represent a place creating tensions between the different users (inhabitants, farmers, industries, naturalists, etc.) [23]. Landscape management has become a key aspect within policy frameworks in the last decades, as attested by, among others, the Pan-European Biological and Landscape Diversity Strategy [24] and the European Landscape Convention [25]. The European Landscape Convention defines landscape as 'an area perceived by people, whose character is the result of the action and interaction of natural and/or human factors'. This emphasizes the key role of human perceptions and values in the definition of landscape and landscape changes.

The concept of ecosystem services (ES) offers a tool that takes a system perspective accounting for the multiple perceptions, values, and benefits of ES providers or beneficiaries [26]. The tool offers a framework that disentangles the complex feedback loops of how management affects ecological processes and ES flows, and how, in turn, these ES changes are perceived [27]. The tool is increasingly used to support decision making on sustainability issues [28], including in contexts of agroecological transitions [29]. Within the proposed framework of Dendoncker et al. [29], assessing the values and perceptions of all stakeholders involved represents a first step to developing a shared understanding of the agro-landscape to further support the co-construction of pathways of change.

While there is a growing body of scientific work being carried out on ES perceptions and values, this seems disconnected from the field of locals' perceptions of agricultural landscape changes. Among the body of literature available regarding the perception of landscape changes [30–32], few studies include the concept of ES explicitly [33]. Previous ES studies have assessed ES perception and values in other ecosystems (e.g., Hicks et al. [34] in coral reefs, Carnol et al. [35] in forests), across various land uses [36–39], for ES identification and selection [40–42], or for participatory mapping [43]. However, few ES perception studies address specific agricultural practices and management regimes [33]. Recently, some exceptions have emerged which address the perceptions of ES delivery within an agricultural context, such as Bernués et al. [33], who focused on animal agriculture, and Andersson et al. [44], who studied intensive and extensive farmlands.

This paper provides a contribution to this vein of work by enquiring of locals (including local inhabitants and conventional and agroecological farmers) about the transitioning aforementioned Hainaut Province about their appreciation and perception of ES delivery in landscapes under agroecological transition. In order to put locals' perception in perspective, we asked 'ES experts' the same questions, and compared these to field-based ES measurements in agroecological and

conventional parcels of the same study region (based on a side study [45]). To sum up, the present research asks four questions: (i) What is the appreciation of locals and experts of landscapes harboring agroecological practices? (ii) What is the perception by locals and experts of ES delivery in these agroecological landscapes? (iii) Do these appreciations and ES perceptions vary between locals and experts? (iv) Are the perceptions of ES delivery similar to that of the ES delivery measured on the field?

The present study will thus provide information on how the landscapes modified by the on-going agroecological transition of the 'innovative farm network' are perceived and appreciated by society. This case study aims at feeding the need to better disentangle the links between agricultural practices, landscape changes, and stakeholders' needs and values.

## 2. Methods

### 2.1. Study Area

The study area was located in the Western part of the Hainaut province in Belgium. This region is located in the 'bas-plateau limoneux Hennuyers' with a topography of plains and low-tablelands where croplands dominate. Small shrub and tree patches are scattered throughout the landscape, with some grasslands near habitations [46]. The study area is representative of intensive agro-landscapes of temperate Western Europe. The climate is oceanic temperate with an annual rainfall of around 800 mm/year and an average yearly temperature of around 10 °C.

Within this landscape dominated by conventional intensive agriculture, a core group of innovative farmers are starting to implement new practices to ensure more autonomy, resilience, and sustainability, creating more diverse and heterogeneous landscapes. Within this 'innovative farms network', some farmers have implemented a whole-system transition. Within these farms, agricultural practices are drastically modified, as they are organically certified, apply reduced tillage to their soil, grow crops in association (referred to as 'intercropping' hereafter), and implement green infrastructures (grass strips, wildflower strips, hedgerows, etc.). By combining all of these agro-ecological practices, we believe that these farms respond to the definition of 'agroecological farming systems' [4,5,10].

### 2.2. Construction of Landscape Scenarios

The approach used in the present analysis takes a system perspective to provide a general picture of how people view transitioning landscapes, taking multiple components and their interactions into account (i.e., we aim at investigating the landscape as a whole, rather than depicting how each landscape element is perceived and appreciated). This approach is relevant in public perceptions of landscapes, as the general public usually sees a landscape as a whole [47]. To do so, people's perceptions were studied through respondents' evaluations of landscape scenarios resulting from photos manipulated with Adobe Photoshop. This approach follows a wide body of studies that successfully assessed people's judgments using manipulated photos as a surrogate for the actual landscape [22,31–33,46]. Scenarios were created from a baseline photograph of a simple landscape of conventional agriculture, representative of the area, but also of the intensive agro-landscapes of Western temperate Europe (Figure 1; conventional landscape (CV)). Creating the scenarios from a baseline photograph decreases potential biases, such as those potentially caused by different weather or landscape structures. The photograph was taken from a dirt road to represent an everyday scene easily experienced by local inhabitants. It was taken with a LUMIX DMC-GF7K on June 16 2016 at 11 a.m., on a sunny and cloudless day. From this photo, agroecological farming practices were added with Adobe Photoshop to construct the agroecological scenario (Figure 1; all agroecological practices (AE)). This scenario combines the following practices: Tree rows to represent agroforestry, wildflower strips, wheat–pulse intercropping, and cattle to represent crop–livestock association systems. To further depict why people's perceptions change between these two contrasted scenarios, the agroecological scenario was de-constructed into each of its different practices, each leading to one scenario. Eventually, this led to six scenarios: The two contrasted scenarios, i.e., the initial conventional landscape (CV) and

the agroecological scenario combining all of the aforementioned agroecological practices (AE), as well as four scenarios depicting a single agroecological practice: Agroforestry (AF), wildflower strips (WF), intercropping (IC), and crop–livestock association (CL) (Figure 1).

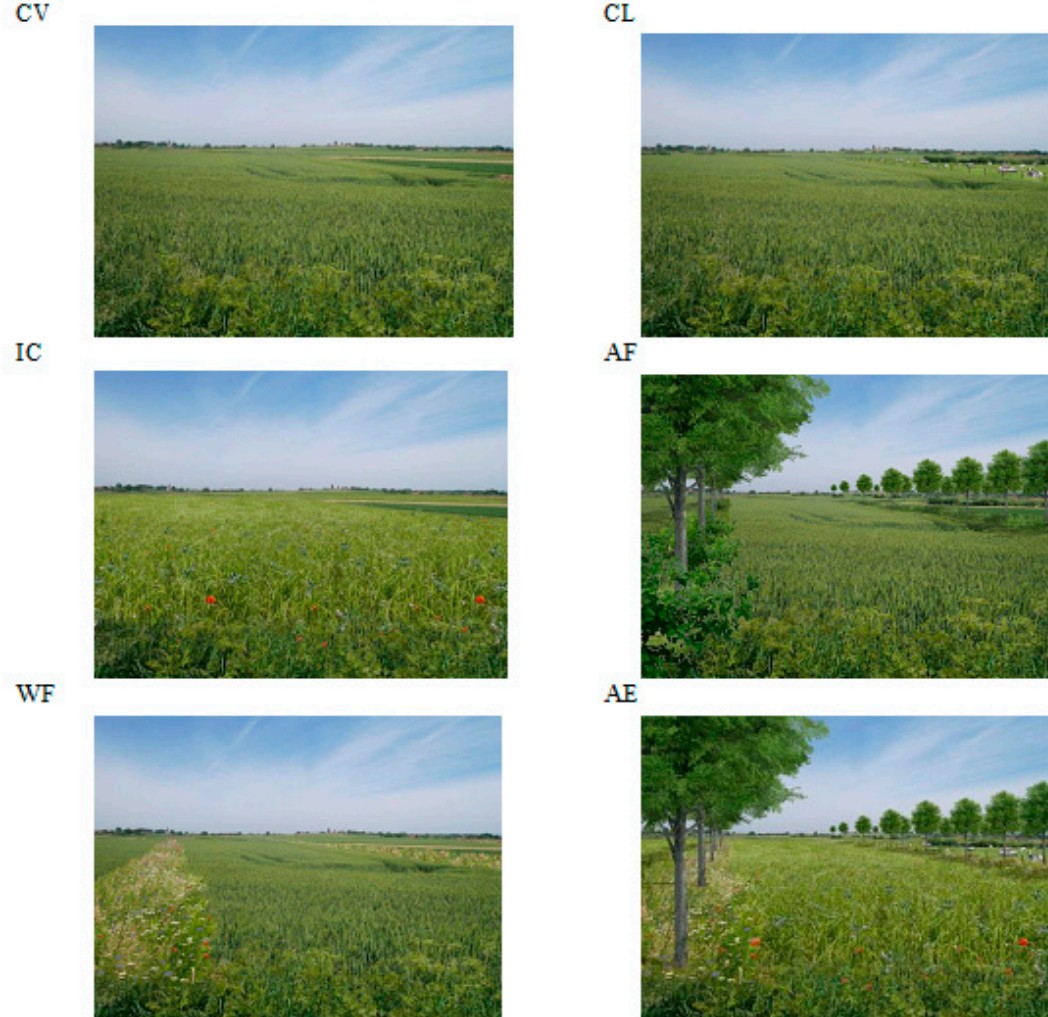

**Figure 1.** Landscape scenarios submitted to respondents for scoring of appreciation and perception of ecosystem service (ES) delivery.

### 2.3. Elicitation of Appreciation and Perception of Ecosystem Service Delivery Across Scenarios

Locals' appreciation and ES perceptions of transitioning agroecological landscapes were elicited by means of a questionnaire asking respondents to score their appreciation and perception of the ES delivery of each of the landscape scenarios described above. In order to put locals' answers into perspective, the same questionnaire was submitted to scientists working on ES (referred to as 'experts' hereafter), and their ES perceptions were compared to the ES field measurements.

Including distinct types of stakeholders (locals and experts) in such perception analyses is important, as different groups can harbor diverging values [48,49]. This can be the underlying cause of unsuccessful policies that are designed by experts while implemented by farmers. The distinction is commonly made between local and scientific knowledge and perception [35,50]. Local knowledge refers to 'knowledge held by a specific group of people about their local ecosystems ( . . . ) derived through various experiential processes ( . . . .), reflects understanding of local phenomena'. On the other hand, scientific knowledge is 'systematic recorded knowledge ( . . . ) passed through a strict and universally accepted set or rules' [50]. In the context of agricultural management, local knowledge is

held by both farmers, who manage the land and influence ES delivery, and local inhabitants, who live in the environment shaped by farmers and benefit from or are impacted by the positive or negative resulting ES flows [34].

The questionnaire was structured around four sections. The first section related to personal data, including sex, age, profession, and the type of living environment. Secondly, the positive and negative feelings regarding the scenarios were enquired through an open question to get insights into participants' mental frameworks without constraints or pre-defined frameworks imposed by scientists [42]. Then, participants were asked to rate the extent to which they believe that each landscape scenario is favorable to the delivery of 13 ESs (ranging from 1: Not at all to 5: Very favorable). The selection of ESs was inspired by a public consultation which took place in March 2015 as an earlier step in the project [42]. The 13 ES included were: Landscape aesthetics, biodiversity, water pollution protection, social cohesion, recreation, pest control, inspiration, heritage, food production, flood protection, erosion protection, and education. The last question of the questionnaire addressed the overall appreciation of each scenario on a 1–5 scale (1: I do not like at all, 5: I like a lot). Such semantic differential scale has been recommended for evaluative approaches [30].

The questionnaire was submitted to locals during a focus group on July 4th 2016, taking place within a wider project that also included a collective valuation that is not presented here. Questionnaires were given at the start of the focus group, before any interaction took place. It is therefore assumed that the focus group setting did not influence the answers provided in the questionnaires. Participants were selected according to a 'purposive sampling' strategy, i.e., sampling in which the participant was selected purposively in order to reach a wide variety of participants, rather than sampling randomly in the population. The aim was to include both ES providers (farmers) and ES beneficiaries (farmers and local inhabitants).

The questionnaire was submitted to ES experts through an online questionnaire sent through the spring 2017 Newsletter of the Belgian Community of Practice on Ecosystem Services [51]. It was specifically mentioned that answers were expected to be the perspective of 'professionals working on ES', in order to distinguish ES expert perceptions from personal perceptions, as the same individual can endorse several roles [27]. The questionnaire was sent on 22 May 2017 and was closed on 28 June 2017. Explanations on the research context, the questionnaire, and what was expected were rigorously similar for both locals and experts.

### 2.4. Biophysical Field-Based Measurements of Ecosystem Services

The ES field measurements took place on agroecological parcels of the innovative farm networks. These parcels harbored the same set of agroecological practices as the ones included in the AE scenario: Agroforestry, wildflower strips, intercropping, and crop–livestock association. The same measurements were also carried out in neighbor parcels with conventional agriculture. Further details on the methodologies and sampling design can be found in [45]. As cultural ESs are intangible and thus difficult to quantify through field measurements, only regulating and provisioning ESs are kept for this part of the analysis.

### 2.5. Statistical Analyses

Statistical analyses were carried out to (i) test within each profile (local or ES expert) whether scenarios were appreciated differently, (ii) test within each profile whether ES delivery was perceived differently across scenarios, and (iii) test whether respondents' profiles led to distinct perceptions of ES delivery or appreciations of the scenario, and (iv) analyze whether locals' perceptions of ES delivery correspond to outcomes of field ES measurements. Since the initial focus of the research was to investigate the perception and appreciation of local stakeholders, and because the profile 'ES expert' serves to put locals' perceptions in perspective, analyses of (i) and (ii) were carried out separately for each profile, even if (iii) did not show significant difference across profiles, in order to provide more detailed analyses.

Analyses were performed in R software version 3.3.2 (R Core Team 2016). Data were tested for normality with Q–Q plots of the residuals. All analyses were carried out using linear mixed models using the package 'lme4' [52]. The respondents' profiles, the scenarios, and the ESs were analyzed as fixed variables, while the individual respondents were analyzed as random variables. Models were constructed from the experimental variables listed above, adding interaction(s) when the Chi-square test (<0.05) using the 'anova' function showed that the interaction significantly affected the model. Multiple comparisons were carried out with the function 'glht' of the 'multcomp' package to depict differences of appreciation between scenarios [53]. The effects of the mixed linear models were tested by means of an F test (<0.05) using the package 'car' [54]. Analyses of the field ES measurements, also using linear mixed models, were done within a side project and are detailed in [45].

## 3. Results

### 3.1. Sample Characteristics

The focus group of locals consisted of 13 participants, including local inhabitants (9) and local farmers (conventional (2) and agroecological (2)). The group was gender balanced (seven males, six females), had a majority of people living in a rural area (ten against two in urban and one in peri-urban areas) and a majority of people aged between 40 and 65 (ten against two between 26 and 39 and one above 65 years old). The questionnaire was answered by 24 ES experts, two-thirds of which were males, and 21 of these persons were aged between 26 and 65 (with only one respondent above 65 and two below 25). The proportion of experts inhabiting rural, urban, and peri-urban areas was evenly shared among respondents (29%, 37%, 33%, respectively).

### 3.2. Appreciation of Agroecological Landscapes

#### 3.2.1. Outcomes of the Scoring Question

Experts and locals do not show significantly different appreciations of the different scenarios ($F_{1,38}$ = 0.434, $P$ = 0.515). Within the profile models, it is shown that experts and locals both appreciate the distinct scenarios differently ($F_{5,24}$ = 12.9, $P$ < 0.001 and $F_{5,13}$ = 8.5, $P$ < 0.001, respectively). Both profiles show the highest appreciation for the agroecological scenario and the lowest for the conventional one (Figure 2) (both $P_{adj}$ < 0.001). Appreciations of the intermediate scenarios do not significantly differ between each other. The conventional scenario is not significantly different from the crop–livestock association for either profile, and, for locals, from the intercropping and the agroforestry scenarios. The agroecological scenario is not significantly different from the wildflower strips, and, for experts, from the agroforestry scenario.

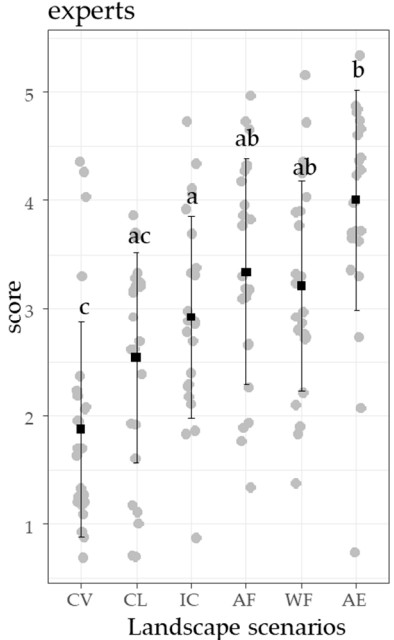
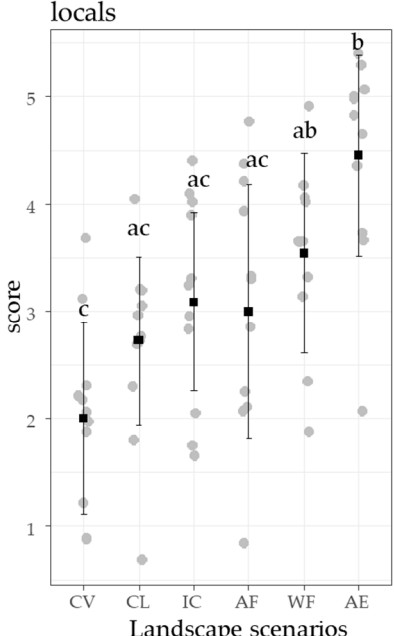

**Figure 2.** Experts' and locals' appreciations of the six scenarios. CV: Conventional, CL: Crop–livestock, IC: Intercropping, AF: Agroforestry, WF: Wildflower strip, AE: Agroecology. Different letters above bars (a, b, c) indicate statistically significant differences between landscape appreciations, while the same letter indicates no significant difference between groups.

### 3.2.2. Outcomes of the Open Question about Positive and Negative Feelings

Results from the open question enquiring about positive and negative feelings regarding scenarios showed several similarities across experts and locals. In general, few comments were directly related to 'feelings' (e.g., comments about the aesthetics or the atmosphere felt); respondents rather commented the structure or function of the agroecosystem with little value judgement (e.g., ES delivered, description of the composition of the agroecosystem). For both groups, many comments were related to biodiversity and diversity in general. Each scenario received both positive (e.g., 'environment more favorable to biodiversity') and negative comments (e.g., 'still not enough habitat diversity to support biodiversity'). Both groups considered all scenarios still too structured and aligned. Within both the expert and the local group, five respondents did not find any positive feelings regarding the conventional scenario, and six respondents did not find any negative feelings regarding the agroecological one.

In addition to these similarities, the results also showed divergences between the perceptions of experts and locals. Overall, experts often mentioned the words 'tranquility/quietness' and 'boring/annoying/dullness', which were never mentioned by locals. The only comments of locals about their feelings referred to a 'sad' landscape, which was mentioned three times. The word 'open' was also thoroughly used by experts (for all scenarios except for the agroforestry and agroecological ones) and was never mentioned by locals. The crop–livestock scenario gathered more negative reactions from experts (e.g., nitrogen deposition, responsibility for climate change, intensive cattle production) and much more enthusiasm from locals (e.g., 'great, cows!', 'nice association between crop and livestock'). However, the main positive aspect of this scenario put forward by experts ('tradition' or 'typical') was not mentioned at all by locals.

Both locals and experts mentioned some negative comments regarding scenarios of isolated agricultural practices, which were not mentioned again for the agroecological scenario. In fact, none of the negative comments mentioned for isolated practices by locals ('low profitability', 'poorly maintained', 'trees too aligned', 'not enough diversity', etc.) were present in the agroecological scenario. For experts, a similar observation can be made for weeds (mentioned in all scenarios except for the

agroecological one) and pesticide use (mentioned in all scenarios except for the agroecological and intercropping one). In the same vein, experts mentioned positive aspects of the conventional scenario that were also present in other scenarios (e.g., 'no construction', 'no allergy').

### 3.3. Perception of Ecosystem Service Delivery in Agroecological Landscapes

#### 3.3.1. Locals' Perceptions of Ecosystem Service Delivery with Regard to Experts' Perceptions

Perceptions of ESs through the distinct scenarios do not differ between experts and locals ($F_{1,5}$ = 0.167, $P$ = 0.685) (Table 1, Figure 3). Within each profile, each ES is perceived as significantly different across the six scenarios, with only one exception: Food production in the eyes of locals ($F_{5,13}$ = 2.22, $P$ = 0.0665). Comparing ES delivery for the agroecological and conventional scenarios reveals that all ESs are perceived as being delivered differently between the two scenarios, both for experts and locals, with the exception of food production ($F_{1,24}$ = 2.42, $P$ = 0.126; $F_{1,13}$ = 0.825, $P$ = 0.375).

**Table 1.** Summary of F and *P*-values of tests run on the different models. The first section provides the outcomes of the model including all variables. Underneath are the results of models per ES run through all scenarios (left), or through the agroecological scenario (AE) and the conventional one (CV) only (right) for both experts and locals. The amount of stars (*) indicates the level of significance for three levels: $P < 0.05$: *; $P < 0.01$: **; $P < 0.001$: ***.

| | Overall Effect | | | | | |
|---|---|---|---|---|---|---|
| | Profile | | Scenario | | ES | |
| | $F_{1,5}$ | $P$ | $F_{5,38}$ | $P$ | $F_{12,38}$ | $P$ |
| | 0.167 | 0.685 | 32.8 | <0.001 *** | 12.1 | <0.001 *** |

| | Difference in ES Perception across all Scenarios | | | | Difference in ES Perception between Scenario AE and CV | | | |
|---|---|---|---|---|---|---|---|---|
| | Experts | | Locals | | Experts | | Locals | |
| | $F_{5,24}$ | $P$ | $F_{5,13}$ | $P$ | $F_{1,24}$ | $P$ | $F_{1,13}$ | $P$ |
| Food production | 4.09 | 0.002 ** | 2.22 | 0.0665 | 2.42 | 0.126 | 0.825 | 0.375 |
| Social cohesion | 17.3 | <0.001 *** | 6.01 | 0.00263 ** | 34 | <0.001 *** | 13.3 | <0.001 *** |
| Soil fertility | 19.9 | <0.001 *** | 8.21 | <0.001 *** | 50.9 | <0.001 *** | 44.1 | <0.001 *** |
| Aesthetic | 28.6 | <0.001 *** | 11.3 | <0.001 *** | 72.8 | <0.001 *** | 43 | <0.001 *** |
| Flood protection | 17.4 | <0.001 *** | 11.3 | <0.001 *** | 37.1 | <0.001 *** | 38.4 | <0.001 *** |
| Heritage | 9.96 | <0.001 *** | 8.34 | <0.001 *** | 23.1 | <0.001 *** | 14 | <0.01 * |
| Erosion protection | 29 | <0.001 *** | 12.8 | <0.001 *** | 55.7 | <0.001 *** | 31.2 | <0.001 *** |
| Recreation | 25.7 | <0.001 *** | 12.1 | <0.001 *** | 49.4 | <0.001 *** | 70.4 | <0.001 *** |
| Biodiversity | 38.1 | <0.001 *** | 25.9 | <0.001 *** | 117 | <0.001 *** | 128 | <0.001 *** |
| Inspiration | 21.4 | <0.001 *** | 12.9 | <0.001 *** | 52.7 | <0.001 *** | 67.2 | <0.001 *** |
| Water poll. prot. | 23.9 | <0.001 *** | 16.6 | <0.001 *** | 52.1 | <0.001 *** | 24.6 | <0.001 *** |
| Education | 22.1 | <0.001 *** | 11.1 | <0.001 *** | 36.6 | <0.001 *** | 52.8 | <0.001 *** |
| Pest control | 38.9 | <0.001 *** | 14.1 | <0.001 *** | 122 | <0.001 *** | 136.1 | <0.001 *** |

Both locals and experts see the agroecological scenario as delivering more ES (Figure 3; light blue) and the conventional scenario as delivering the least ES (Figure 3; green). The intermediary scenarios follow the same trend for both profiles: The crop–livestock association is perceived as delivering less ES, followed by the intercropping scenario. The distinction between perceived ES delivery of the wildflower strip and the agroforestry scenario is not significant.

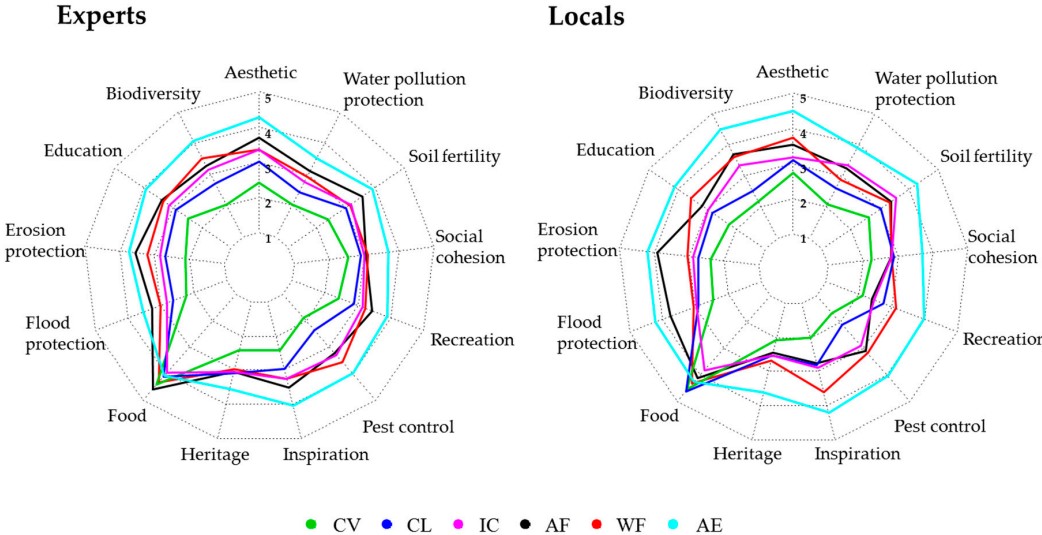

**Figure 3.** Radar plot of the average perceptions of ES delivery for experts and locals. CV: Conventional, CL: Crop–livestock, IC: Intercropping, AF: Agroforestry, WF: Wildflower strip, AE: Agroecology.

Appreciation of the different scenarios followed the same trends for both locals and experts. Furthermore, these appreciations also follow the trends of ES delivery perceptions (Figure 4).

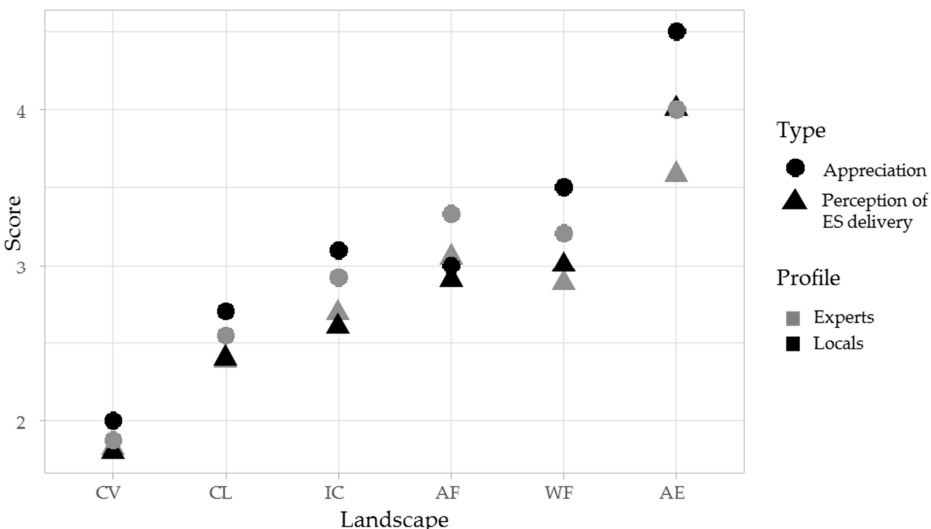

**Figure 4.** Average appreciation and perception of ES delivery for experts and locals. CV: Conventional, CL: Crop–livestock, IC: Intercropping, AF: Agroforestry, WF: Wildflower strip, AE: Agroecology.

### 3.3.2. Locals' Perceptions with Regard to Field ES Measurements

As shown above, locals tend to perceive as much food production in the AE and the CV scenarios, but more regulating ESs in AE. These trends correspond partially to the field ES measurements (Table 2): One out of two food production indicators (straw yield) is quantified as similar (i.e., not significantly different) in the AE and CV parcels, and three out of nine regulating ES indicators are significantly higher in the AE parcels (erosion protection, soil respiration rate, and aphid control) [45]. Thus, similar trends are observed between locals' perceptions and ES field measurements: The productivity seems to be only partially impacted by AE practices, and more regulating ESs were measured in AE parcels, as was perceived by locals.

**Table 2.** ES delivery in agroecological (AE) and conventional (CV) systems as perceived by locals, based on manipulated landscape photographs, and as quantified through field measurements. For more information on the field ES measurements, the indicator chosen, and the methods followed, see [45].

| | ES Delivery: AE vs. CV | | | | | | |
|---|---|---|---|---|---|---|---|
| | **Locals' Perception** | | | **Field-Based ES Measurements** | | | |
| | $F_{1,13}$ | *P* | Outcomes | Indicator | $F_{1,78}$ | *P* | Outcomes |
| Food production | 0.825 | 0.375 | AE = CV | Straw yield (kg/m$^2$) | 0.01 | 0.93 | AE = CV |
| | | | | Grain yield (kg/4 m$^2$) | 141 | <0.001 | CV > AE |
| Soil fertility | 44.1 | <0.001 | AE > CV | Soil organic matter degradation rate (%) | 1.9 | 0.302 | AE = CV |
| | | | | Soil respiration rate (mgCO$_2$/g) | 74.5 | <0.001 | AE > CV |
| | | | | Sum of nutrients (g/kg) | 0.004 | 0.9489 | AE = CV |
| Flood protection | 38.4 | <0.001 | AE > CV | Soil permeability (cm/day) | 0.552 | 0.459 | AE = CV |
| Erosion protection | 31.2 | <0.001 | AE > CV | Soil aggregate stability (0–5 class) | 18.3 | 00.0433 | AE > CV |
| Water poll. prot. | 24.6 | <0.001 | AE > CV | Potentially leaching Nitrogen (Kg N-NO$_3$/ha) | 1.34 | 0.258 | AE = CV |
| Pest control | 136.1 | <0.001 | AE > CV | Parasitism rate (%) | 0.302 | 0.592 | AE = CV |
| | | | | Aphid abundance | 25.8 | <0.001 | AE > CV |
| | | | | Predation rate (%) | 0.12 | 0.731 | AE = CV |

## 4. Discussion

In spite of the limited sample size, our study presents some clear trends: From the scenario photographs, landscape changes induced by agroecological transitions are perceived positively by the local population. They are better appreciated, and are perceived as delivering more regulating ESs and as being as productive as conventional landscapes. Experts' perceptions and appreciations follow the same trends as those of the locals, indicating a shared understanding of the complex interactions between agricultural practices, landscape modification, and ES delivery. These trends correspond to field ES measurements carried out in agroecological and conventional parcels of the same study area. While the 'yield gap' is often put forward as the major challenge faced by agroecological practices, this does not seem to be what is perceived by locals of this area undergoing agroecological transition. This section first discusses the study limitations, and then the positive perception of agroecology; we eventually reflect on how our results question this focus on the 'yield gap' and how we suggest future research to foster agroecological transition.

### 4.1. Study Limitations

Although our results show clear and statistically significant outcomes, the study relies on a small sample size, and the selection of the locals may be biased towards people sensitive to the question of sustainable agriculture and landscapes. This study does not aim to represent the global rural population, as it focuses on a specific local context. Local-based approaches are relevant when addressing landscape perceptions, since preferences for landscape attributes are highly context-specific [17,18,55]. Similar research aiming to support rural landscape management should broaden the population sample to reach higher representativeness. Additionally, the appreciations and perceptions of the landscapes were based on scenarios constructed from manipulated photographs. Our results are thus to be interpreted in terms of perceptions and appreciations of *agroecological-like* scenarios. Thus, this represents an indirect link to real-life agroecological landscapes or parcels used for the field measurements, or to the concept of agroecology itself (there were no explicit reference to the terms 'agroecology' or 'agroecological practices'). At last, when comparing locals' and experts' appreciations through the open question, it is unknown whether differences are due to a difference in the appreciation per se or to the use of distinct vocabulary.

### 4.2. Agroecology Perceived as a Synergetic Whole

Our results suggest a general positive a priori with regard to agroecology and diversified farming landscapes. Indeed, these were better appreciated and were seen as productive as conventional ones and as delivering greater regulating ESs. Negative comments formulated with regard to isolated practices disappeared when combined in the agroecological scenario, illustrating how agroecology is seen as a synergetic whole. Scenarios perceived as delivering more ESs were also more appreciated, as corroborated by earlier work, in which multifunctional landscapes providing a wide array of ESs were preferred [36] and linked to increased wellbeing [56]. In this vein of work, previous studies have identified that more appreciated landscapes relate with landscapes involving fruitful practices, fertility indicators, or other symbols of sustainable human subsistence [57,58].

These appreciations and perceptions of ES delivery showed similar trends for both locals and experts. Such similarities have been shown in previous research, where farmers' perceptions were similar to those of conservationists [33] or scientific literature [49]. Other studies in the same direction do not abound, and depict different perceptions between rural communities and scientists or conservationists [27,39]. In fact, results could possibly not be consistent across studies, as the attitudes and perceptions of locals and the interactions between agricultural practices and ES delivery all vary with their contexts [59].

### 4.3. Agroecology Multi-Performance Assessment: Bridging the ES Gap

In order to put into perspective these positive perceptions of agroecology with regards to what is actually happening in agroecological fields, these were compared to field ES measurements in agroecological and conventional parcels of the same study area [45]. These perceptions of delivery partially correspond to field ES measurements, with more regulating ESs being delivered by agroecology and with a similar productivity for straw yields.

Other studies have shown that the tradeoff between agricultural production and other ES delivery does not exist in the view of locals and farmers [49]. In fact, recent quantitative research shows that agroecology can conciliate food provision with bundles of other ESs [60–64]. In the light of these successful examples and of such positive perceptions by locals and farmers, we concur with Rapidel et al. [65] to focus on the 'multi-performance' of agroecosystems with the aim of bridging 'the ES gap' instead of focusing only on the productivity and the 'yield gap'. As yields of intensive agriculture come at the cost of destroying ecological processes, which, in turn, impacts crop development, all ESs should be included in the assessment of agricultural system performance [2].

### 4.4. Research Avenues to Foster Agroecological Transitions

In the present research, three ES valuation approaches were analyzed concurrently: Locals' ES perceptions, experts' ES perceptions, and biophysical field-based ES measurements. Distinct valuation approaches bring complementary pieces of information: Locals' knowledge can better embrace the values and stakes involved, while expert knowledge can complement with scientific expertise and bring scientific aspects to the front which are not visible to the broad public [42]. On the other hand, biophysical field measurement showed yet another prism of analysis, with similar trends, but differences in specific ESs. Within the biophysical approach itself, different indicators or measurement methods for the same ES led to distinct ES outcomes ([45], Table 5). At the end, each approach, be it social or biophysical, informs on one of the facets of the inherent complexity of a transitioning agroecosystem [66]. Rather than claiming that one of the approaches holds better scientific wisdom or better assesses 'the reality', we believe that each approach brings a different and complementary piece of information to reveal a 'subjective reality' [67].

Considering the inherent complexity of agricultural transitioning systems and the involvement of a great diversity of actors (from coproducers and managers of ES to ES beneficiaries), it has been increasingly suggested to move to a new valuation school of 'integrated valuation' when aiming at assessing or steering (agricultural) transitioning systems [29,68]. This requires integrating diverse valuation approaches to integrate both local and scientific knowledge [69,70]. Integrated valuations take part in the broader category of 'mixed methods research', an innovative methodological approach combining qualitative and quantitative data and methods within a single research [71]. The complexity of research tackling sustainability and transition issues calls for answers beyond simple numbers or words. It is thus believed that a combination of different forms of data can provide the most complete analysis to support the co-design of sustainable rural landscapes relevant to their socio-ecological contexts [36,49].

## 5. Conclusions

In literature, calls for research on agricultural transition through the prism of the concept of ESs abound. While research studying the perception of landscape change expands, the integration of the ES concept within this vein of work remains weakly explored. Our study provides a snapshot assessment by analyzing how locals and experts appreciate landscapes undergoing agricultural transitions and how these are perceived in terms of ES delivery. By deconstructing an agroecological scenario into its individual practices, the study showed that the agroecological scenario was better appreciated and was perceived as delivering more ESs. Furthermore, negative feelings arising for isolated practices disappeared when combined in the scenario of agroecology. Our results thus suggest a positive a priori

of agroecological landscapes both in terms of general appreciation and ES delivery. These positive reactions were similar between locals and ES experts, while ES delivery measured quantitatively based on field measurements corresponded to those perceptions.

Farmers are ES providers, but are also impacted (positively or negatively) by ES flows. Thus, implementing their knowledge into rural management is likely to bring complementary insight to traditional scientific approaches. By being locally specific and time-specific, we believe that local knowledge can contribute to feeding and complementing expert knowledge and biophysical approaches to lead to a more complete analysis of transitioning systems which are complex, do not follow linear trends, harbor stakes, and are context-specific. To do so, such research ought to be embedded within a wider iterative framework, as suggested by Dendoncker et al. [28], in which the understanding of the broad set of values and perceptions of all of the stakeholders involved allows co-designing and exploring of the potential evolutions of the agro-landscape and the selection of the most acceptable and socially and environmentally sustainable pathway of change.

**Author Contributions:** Conceptualization, G.M.; Formal analysis, F.B.; Funding acquisition, M.D., N.D. and G.M.; Investigation, F.B. and A.D.; Methodology, F.B., A.D. and G.M.; Software, A.D.; Supervision, G.M.; Writing—original draft, F.B.; Writing—review and editing, M.D., N.D., and G.M. All authors have read and agreed to the published version of the manuscript.

**Funding:** This research was funded by the National Research Fund for Research of Belgium (Fond National de la Recherche Scientifique—FNRS).

**Acknowledgments:** We wish to thank all participants in the focus groups and the questionnaire, without whom the presented work could not have taken place. We thank the Parc Naturel des Plaines de l'Escaut for their support in reaching participants for the focus group and for providing the material and technical support for the organization of the meeting. We thank the Fond National de la Recherche Scientifique (FNRS) for the financial support of this research.

**Conflicts of Interest:** The authors declare no conflict of interest.

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
