# Peer review of "How Are Landscapes under Agroecological Transition Perceived and Appreciated? A Belgian Case Study"

_sustainability, doi:10.3390/su12062480_

Round 1

Reviewer 1 Report

REVIEWER 4 - revision

I appreciate the efforts of Boeraeve and colleagues aimed to improve their ms. “How Are Landscapes under Agroecological Transition Perceived and Appreciated? A Belgian Case Study”. In my view it was mostly a successful endeavour. The major asset to the study was enriching it with the postulated (by myself as well as by the Rev. 2) quantitative ecological data. They provide a sound perspective, to which the perception data can be referred. Alas, considering the narrative line, the Authors seem undervalue their own work, suggesting a non-scientific, rather ideological construct of an “integrated valuation”, which is not a synonym of a highly appreciated multi-disciplinary approach.

My final advice to the Authors is considering a (minor) revision that would involve the suggested changes. Some of them I consider essential from a point of view of the philosophy of science (should we stick to the rigour and clarity of the scientific methodology, language, discipline, or follow ideological obscure discourses?). The advised changes would not require any ‘earthquake’ to the study - just some rephrasing. Other changes are necessary because of the obvious errors in the results presentation (see the following detailed remarks).

Dialogue with the Authors

>>(...) Regarding the point that ‘environmental sustainability’ should only be evaluated through ‘well-calibrated best expert knowledge’, we however do not share entirely this point of view. Literature on sustainability abounds to emphasize the need to further explore how combining ecological, socio-cultural valuation tools can support resource and land use decision-making (…)”

<<Not only. In particular, through quantitative assessments of parameters characterizing a landscape's performance, resilience, etc. I do not deny that there are numerous references on the "sustainability" issue. This issue is certainly "complex" or rather hardly defined. Therefore, its use in a scientific narrative is questionable. It may be useful in an ideological prattle but should be avoided in science.

>> We thank the reviewer for pointing to this lack of clarity. We have now rephrased as follows: ‘This definition emphasizes the key role of human perceptions and values as the drivers of landscape changes’

<<Actually, it does not. If the flight of a bat is perceived by people, it does not mean that the human perception is a driver of that flight in any way. The same applies to landscapes unless we consider consciously designed landscapes by landscape architects...

>> We agree that the link can be indirect but our aim was not to quantify specific links between specific practices and specific ES, but rather to take a holistic system perspective as now specified in the paper: ‘The approach used in the present analysis takes a holistic system perspective to provide a general picture of how people view transitioning landscape, taking multiple components and their interactions into account. Such approach is relevant in public perceptions of landscapes as the general public usually sees landscape as a whole [47].’

<<I am sorry, but "holistic approach" (also extensively used in the Authors' response to Reviewer 2) evokes somehow "Chinese medicine". Considering multiple factors, variables is not a sin, but replacing the clarity of clear-cut definitions and parameters with esoteric concepts is not a good idea.

>>We do not believe that the results are ‘expected’: the interviewees included farmers farming with conventional and rather intensive practices which could have been expected to perceive agroecological practices as hindrances and less productive.

<<But the questions were not about productivity. I do not presume that "conventional farmers" were not able do distinguish "ecological sustainability" (whatever it is...) from the economic suitability. If they were asked about possible impacts of the "transition agroecology" on the farming economy - you would certainly get a far more diverse set of answers.

Manuscript - detailed remarks

L55 (and further down): “feed autonomy” - shouldn’t be “food autonomy”?

L72-73: “This definition emphasizes the key role of human perceptions and values as the

drivers of landscape changes.”

<< Actually, it does not. If the flight of a bat is perceived by people, it does not mean that the human perception is a driver of that flight in any way. The same applies to landscapes unless we consider consciously designed landscapes by landscape architects...

L101, 201, 358-359: “corroborate”

<< perhaps “correspond to” would be better? More importantly, biophysical measurements (providing information on objective facts) can “corroborate” perceptions (subjective feelings), not the other way around...

L171: “social cohesion”

<<How "social cohesion" can be elicited from pictures, which do not encompass any aspect of a community performance: no houses, no church on the horizon... It refers to some imprinted ideological shortcuts rather to genuine knowledge, experience, or even feeling...

L218-225: “The focus group (...) 33% respectively).”

<<Do such detailed facts about small samples make any sense unless one compares the preferences of olds to youngs, men to women?

L234-237: “The agroecological scenario is not significantly different from the crop-livestock association for both profile, and for locals, also from the intercropping and the agroforestry scenarios.

<<The figure shows the opposite: AE differs from CL for both profiles!

“The conventional scenario is not significantly different from the wildflower strips, and for experts also from the agroforestry scenario.”

<<For experts CV does differ from all other scenarios except for CL!

L254-255: “Overall, experts often mentioned the words ‘tranquility/quietness’ and ‘boring/annoying/dullness’”

<<This may say something their professional expertise or just about a bit richer vocabulary?

L256: “sad” - in fact this sounds more genuine, authentic than the experts’ cliches...

L258-259: “The crop-livestock scenario gathered more negative reactions from experts (e.g. nitrogen deposition, responsible of climate change, intensive cattle production)”

<<How did they infer from such a picture that it was "intensive", contributing to climate change?

L261-262: “(...) experts (‘tradition’ or ‘typical’) was not mentioned at all by locals.”

<<Possibly, because it was neither typical or traditional...

Figure 4

<<Legend: in 'profile' there should be a neutral shape, e.g. a rectangle, used to depict to colours, instead of circle, which illustrates appreciation only.

L296-297 - is the sentence grammatically correct?

L371-372: “In the present research, three ES valuation approaches were put in perspective: locals’ ES perception, experts’ ES perception, and biophysical field ES measurements.”

<<Isn't it some newspeach? Put in perspective of what? I would suggest changing this to something like: "In the present research has put the ES perceptions and appreciations in perspective of the biophysical field ES measurements. It has also compared the local farmers' feelings to experts' subjective assessments regarding the four different scenarios of the agricultural landscape."

L375-382: “On the other hand, biophysical field measurement showed yet another prism of analysis, with similar trends but differences on specific ES. Within the biophysical approach itself, different indicators or measurement methods for a same ES led to distinct ES outcomes ([45], Table 5). At the end, each approach, it being social or biophysical, informs on one of the facets of the inherent complexity of a transitioning agroecosystems [66]. Rather than claiming that one of the approaches holds better scientific wisdom, or better assesses ‘the reality’, we believe each approach brings a different and complementary piece of information to reveal a ‘subjective reality’ [67].”

L385-388: (...) ‘integrated valuation’ when aiming at assessing or steering (agricultural) transitioning systems [29,68]. This requires integrating diverse valuation approaches to integrate both local and scientific knowledge [69,70]. Integrated valuations take part to the broader category of ‘mixed methods research’,”

<<I would suggest the Authors emphasizing the high correspondence (in this case!) of the perception with the objective empirical data rather than promoting all approaches as equally valuable sources of "wisdom". Although I accept that the traditional ecological knowledge (locally-specific) is an important source of wisdom (accumulated by experiences of the farmers' generations), independent from regular research, I do not think 'experts' knowledge' having a similar quality. It is rather a (very) generalized research knowledge (not necessarily best applicable to a certain locality).

L375: “not be visible”

L373: “THE locals’ knowledge”

L388: “AN INnovative” (or “a novel”)

Reviewer 2 Report

The work is overall quite interesting, and for sure it will deserve some room in the journal. I would appreciate a better description of the research design, and a more precise abstract in order to immediately clarify the methods of the research. 

Reviewer 3 Report

I'd like to thanks the authors for a much improved manuscript, and for addressing the comments raised in earlier comments. My final suggestion is a minor one, easily done, the radar plots (Fig 3) are easier to interpret if the points are joined. The colours used are also hard to distinguish , especially if readers are colour blind, so have another look.

Author Response

>> We thank the reviewer 4 for his recommendations. We have added lines between dots to improve readability on Figure 3. We could however not find a set of colors more contrasted than the one used presently. A solution to have the graph readable in black and white prints would have been to use distinct line types but this brings confusion as they are too many of them.  We have thus kept the graph as such, but we believe that the use of lines between dots will improve readability substantially.

Reviewer 4 Report

Changes were done in a properly way

Author Response

>>> We thank the reviewer for his second review and making sure that modifications were brought in a proper way.